# Practical Application of the Galerkin Finite Element Method with a Mass Conservation Scheme under Dirichlet Boundary Conditions to Solve Groundwater Problems

**Heejun Suk [1], Jui-Sheng Chen [2],\*, Eungyu Park [3] and You Hong Kihm [1]**

[1] Korea Institute of Geoscience and Mineral Resources, Daejeon 34132, Korea; sxh60@kigam.re.kr (H.S.); kihmyh@kigam.re.kr (Y.H.K.)
[2] Graduate Institute of Applied Geology, National Central University, Taoyuan City 320, Taiwan
[3] Department of Geology, Kyungpook National University, 80 Daehak-ro, Buk-gu, Daegu 41566, Korea; park.eungyu@gmail.com
\* Correspondence: jschen@geo.ncu.edu.tw; Tel.: +886-3-2807427

**Abstract:** The Galerkin finite element method (FEM) has long been used to solve groundwater flow equations and compute the mass balance in a region. In this study, we proposed a simple, new computational FEM procedure for global mass balance computations that can simultaneously obtain boundary fluxes at Dirichlet boundary nodes and finite element hydraulic heads at all nodes in only one step, whereas previous approaches usually require two steps. In previous approaches, the first step obtains the Galerkin finite element hydraulic heads at all nodes, and then, the boundary fluxes are calculated using the obtained Galerkin finite element hydraulic heads in a second step. Comparisons between the new approach proposed in this study and previous approaches, such as Yeh's approach and a conventional differential approach, were performed using two practical groundwater problems to illustrate the improved accuracy and efficiency of the new approach when computing the global mass balance or boundary fluxes. From the results of the numerical experiments, it can be concluded that the new approach provides a more efficient mass balance computation scheme and a much more accurate mass balance computation compared to previous approaches that have been widely used in commercial and public groundwater software.

**Keywords:** Galerkin finite element method; global mass balance; Dirichlet boundary; boundary flux

## 1. Introduction

The Galerkin finite element method (FEM) has long been used to solve groundwater flow and advection–dispersion–reaction equations to predict groundwater flow and the transport of pollutants in porous media. Popular commercial simulation programs, such as FEMWATER [1], FEFLOW [2], and HYDRUS3D [3], were developed based on the Galerkin FEM, and programs such as these have been widely used for some time. In these commercial software packages, Galerkin FEM is used to solve the governing equation of groundwater flow subject to appropriate boundary and initial conditions. The governing equation is simply a statement of a water mass conservation equation coupled with constitutive relations, such as Darcy's law. In this conventional FEM formulation, the pressure or hydraulic head distribution is obtained and a velocity field is subsequently calculated using Darcy's law by taking the derivatives of the calculated pressure head distribution, which is used as either the advection velocity for calculating the contaminant transport or the flow through a boundary for calculating the water mass balance. This approach toward obtaining the velocity field is denoted here

as the conventional differential approach (CDA). However, the calculated velocity field using the CDA often contains velocity discontinuities at nodal points and element boundaries. Such discontinuities unfortunately lead to large errors when solving the contaminant transport equation. In addition, discontinuities can also lead to failure to conserve the water mass in mass balance computations. The CDA was included in the HYDRUS software.

Yeh [4] demonstrated water balance errors in the range of 24–30% for a complex problem due to discontinuities in the computed Darcy flux in the interior of the domain. An alternative postprocessing approach was proposed, which provides a continuous Darcy flux by applying the finite element approach used to simulate the groundwater head field to the Darcy equation, with the fluxes as the state variables. Yeh reported that the global mass balance errors could be reduced from 23.8% to 2.2% when this postprocessing approach was used rather than the CDA. Yeh's postprocessing approach is already included in the FEMWATER software and several other studies have applied it to a range of groundwater flow and transport simulation problems [5–10].

Lynch [11] showed that a precise global mass balance can be achieved via the Galerkin FEM by focusing only on calculating the boundary flux at a Dirichlet boundary rather than calculating a continuous Darcy flux over a whole domain. It was shown through mathematical analysis that the common practice of discarding the Galerkin equations violates the mass balance by requiring that these fluxes be approximated. In contrast, by retaining the Galerkin equations at Dirichlet boundaries as the equations for the boundary flux, a precise global mass balance was demonstrated through conceptual mathematical and hypothetical abstract examples. By retaining the Galerkin equations at Dirichlet boundaries as the equations for the boundary flux, Carey [12] showed that boundary fluxes can be calculated with exceptional accuracy. He demonstrated from numerical studies that the boundary flux errors will be $O(\Delta x^{2k})$, where $k$ is the degree of the element polynomial basis if the exact solution is sufficiently smooth. Accordingly, he concluded that the calculated boundary fluxes not only have exceptional accuracy but also higher rates of convergence compared to the calculated fluxes using the CDA. It has also been observed by other researchers that the postprocessing technique suggested by Lynch [11] provides very accurate mass balances [12–16]. However, in all of these studies, the advantages of the postprocessing technique were demonstrated only through a conceptual mathematical framework, which was too simple or used hypothetical abstract examples that were far from typical groundwater scenarios or practical application problems. Most examples were limited to a one-dimensional steady state with a homogeneous material and simple boundary conditions or a simple geometry. However, typical groundwater problems are characterized by multi-dimensional, heterogeneous, and transient features, as well as various source/sinks and a complex geometry. Furthermore, in all previous approaches using either Yeh's postprocessing technique or the CDA, two steps are usually needed to calculate the boundary flux at a Dirichlet boundary. In the first step, Galerkin finite element solutions are obtained by solving an algebraic matrix equation, and then, in the next step, the boundary flux is calculated. Even Lynch's approach calculates integral boundary fluxes by substituting the obtained finite element solutions into the retained Galerkin equations at Dirichlet boundaries after solving for the finite element pressure or hydraulic head distributions, and hence, also requires two steps.

Furthermore, the same idea of Lynch [11] and Carey [12] can be extended to compute not only boundary fluxes but also the internal fluxes [14,16]. For the computation of the internal fluxes, the Galerkin equations at the interior nodes will be retained, and then, by treating that node as a Dirichlet boundary, the internal flux can be solved with the Galerkin equation at the node using the groundwater head at an internal node computed with the Galerkin FEM. The need to calculate the internal flux often arises when detailed inflow/outflow components are to be examined at the subdomain level during the calibration and verification phase of modeling studies. On the other hand, an alternative postprocessing method that calculates the internal flux was developed by assuming that the flow field is irrotational [17–20]. The alternative postprocessing method subdivides elements into patches and individual fluxes for each patch are computed to calculate flow rates through each of the

element faces such that flow through the boundary of any subdomain can be calculated by summing the flow rates at those faces that define the boundary. In this study, we focussed on the calculation of only the boundary fluxes to calculate the global mass balance, rather than the internal fluxes.

In this study, a new and simple computational procedure incorporating the postprocessing approach described by Lynch [11] was proposed to simultaneously obtain boundary fluxes at the Dirichlet boundary nodes and finite element hydraulic heads at all nodes in only one step. The proposed procedure was applied to typical groundwater scenario examples to illustrate its applicability to realistic groundwater problems. Furthermore, a comparison between the postprocessing approach described by Yeh [4], the conventional differential approach (CDA), and the new approach proposed in this study was performed using two practical groundwater problems to illustrate the accuracy and efficiency of the new approach for computing the global mass balance or boundary fluxes.

## 2. Methodology

In this study, a new computational procedure was introduced based on the postprocessing approach described by Lynch [11]. In the new approach, the global matrix and load vectors are assembled in the Galerkin FEM and the integral boundary fluxes at Dirichlet boundary nodes are assigned as primary variables to be solved, as well as the hydraulic heads at all nodes, except the Dirichlet nodes. Accordingly, the integral boundary fluxes at the Dirichlet nodes and the finite element hydraulic heads at all nodes, except the Dirichlet nodes, can be solved using only one step. The governing equation of water flow in a saturated–unsaturated porous medium can be written as follows [1,21–23]:

$$F\frac{\partial h}{\partial t} - \nabla\cdot(K_s k_r \cdot \nabla H) - Q = 0, \tag{1}$$

where $F = \frac{\theta}{n_e}\alpha' + \beta' + \frac{d\theta}{dh}$; $h$ is the pressure head; $H = h + z$ is the total head; $\theta$ is the moisture content; $n_e$ is the effective porosity; $\alpha'$ and $\beta'$ are the modified coefficients of the compressibility of the medium frame and the water, respectively; $K_s$ is the saturated hydraulic conductivity; $k_r$ is the relative permeability; $Q$ is the source or sink; $z$ is the vertical coordinate (positive upward); and $t$ is time. To solve the nonlinear flow equation (Equation (1)), constitutive relations must be established that relate the primary unknown $h$ to the secondary variables $\theta$ and $k_r$. In this study, without the loss of generality, Gardner constitutive relations were used to solve the transient flow in variably saturated porous media. In Gardner constitutive relations, the water content and relative permeability are given as simple exponential functions of the pressure head, as follows:

$$\theta = \begin{cases} \theta_r + (\theta_s - \theta_r)e^{\lambda h} & \text{for } h < 0, \\ \theta_s & \text{for } h \geq 0, \end{cases} \tag{2}$$

$$k_r = \begin{cases} e^{\lambda h} & \text{for } h < 0, \\ 1 & \text{for } h \geq 0, \end{cases} \tag{3}$$

where $\theta_s$, $\theta_r$, and $\lambda$ represent the saturated water content, the residual water content, and a soil index parameter related to the pore-size distribution, respectively. The Darcy velocity can be calculated using:

$$V = -K\cdot\nabla H, \tag{4}$$

where $K = K_s k_r$ is the hydraulic conductivity. The initial condition can be written as:

$$h = h_0(x) \text{ in } R, \tag{5}$$

where $h_0$ is a prescribed function of the spatial coordinate $x$, and $R$ is the region of interest. In the Galerkin FEM, the weighted residual integral equation can be written using weighting functions $N_i$, as follows:

$$\int_R N_i \left\{ F \frac{\partial \hat{h}}{\partial t} - \nabla \cdot \left[ K \cdot \nabla \left( \hat{h} + z \right) \right] - Q \right\} dR = 0, \qquad i \in N, \tag{6}$$

where $N_i$ is the weighting function at node $i$, $\hat{h}$ is a trial function of $h$, and $N$ is the total number of nodes in the finite element network. The trial function $\hat{h}$ can be calculated using:

$$\hat{h} = \sum_{j=1}^{N} N_j(x) h_j, \tag{7}$$

where $h_j$ is the hydraulic head at node $j$. Using Green's theorem to remove the second derivative and substituting Equation (7) into Equation (6), one obtains the following:

$$\sum_{j=1}^{N} \int_R N_i F N_j dR \frac{\partial h_j}{\partial t} - \int_B N_i \hat{n} \cdot \left[ K \cdot \nabla \left( \hat{h} + z \right) \right] dB$$

$$+ \sum_{j=1}^{N} \int_R \nabla N_i \cdot K \cdot \nabla N_j dR h_j + \int_R \nabla N_i \cdot K \cdot \nabla z \, dR$$

$$- \int_R N_i Q dR = 0, \; i \in N, \tag{8}$$

where $B$ is the boundary of the solution region and $\hat{n}$ is the outward unit vector normal to $B$. The resulting system of nonlinear ordinary differential equations (Equation (8)) can be solved in time using the backward (implicit) Euler finite difference scheme. Accordingly, the final nonlinear system can be written by substituting Equation (4) into Equation (8), as follows:

$$\sum_{j=1}^{N} \int_R N_i F^{n+1} N_j dR \frac{h_j^{n+1}}{\Delta t} + \sum_{j=1}^{N} \int_R \nabla N_i \cdot K^{n+1} \cdot \nabla N_j dR \, h_j^{n+1}$$

$$= \sum_{j=1}^{N} \int_R N_i F^{n+1} N_j dR \frac{h_j^n}{\Delta t} - \int_R \nabla N_i \cdot K^{n+1} \cdot \nabla z \, dR + \int_R N_i Q^{n+1} dR$$

$$- \int_B N_i \hat{n} \cdot V^{n+1} dB, \; i \in N, \tag{9}$$

where the superscripts $n+1$ and $n$ are the new and old time levels, respectively, and $\Delta t$ is the time step. These equations can be conveniently written in matrix form as follows:

$$\mathbf{M}^{n+1} \cdot \mathbf{h}^{n+1} + \mathbf{S}^{n+1} \cdot \mathbf{h}^{n+1} = \mathbf{M}^{n+1} \cdot \mathbf{h}^{n} - \mathbf{d}^{n+1} - \mathbf{b}^{n+1}, \tag{10}$$

where

$$\left[ M_{i,j}^{n+1} \right] = \int_R N_i F^{n+1} N_j dR \frac{1}{\Delta t}, \qquad i, j \in N, \tag{11}$$

$$\left[ S_{i,j}^{n+1} \right] = \int_R \nabla N_i \cdot K^{n+1} \cdot \nabla N_j dR, \qquad i, j \in N, \tag{12}$$

$$\left\{ d_i^{n+1} \right\} = \int_R \nabla N_i \cdot K^{n+1} \cdot \nabla z dR - \int_R N_i Q^{n+1} dR, \qquad i \in N, \tag{13}$$

$$\left\{ b_i^{n+1} \right\} = \int_B N_i \hat{n} \cdot V^{n+1} dB, \qquad i \in N. \tag{14}$$

It should be noted that $\{b_i^{n+1}\}$ is denoted by Lynch [11] as the Galerkin flux or integral boundary flux at boundary node $i$ and new time level $n+1$. Lynch demonstrated that if boundary node $i$ is a Dirichlet boundary node, the conventional practice of eliminating the Galerkin equations in Equation (10) at Dirichlet boundaries destroys the mass balance because the flux at the boundary should instead be calculated as the gradient of the obtained pressure or hydraulic head using Darcy's law. Accordingly, he suggested that the Galerkin equations at Dirichlet boundaries should be retained as the algebraic equations for the boundary flux to ensure a good mass balance. Therefore, Equation (10) can be rewritten as:

$$\mathbf{A^{n+1}} \cdot \mathbf{h^{n+1}} = \mathbf{c^{n+1}} - \mathbf{b^{n+1}}, \tag{15}$$

where:

$$\mathbf{A^{n+1}} = \mathbf{M^{n+1}} + \mathbf{S^{n+1}}, \tag{16}$$

$$\mathbf{c^{n+1}} = \mathbf{M^{n+1}} \cdot \mathbf{h^n} - \mathbf{d^{n+1}}. \tag{17}$$

Expressing Equation (15) as a matrix and a load vector gives the following:

$$
\begin{bmatrix}
A_{1,1}^{n+1} & A_{1,2}^{n+1} & A_{1,3}^{n+1} & \cdots & A_{1,N}^{n+1} \\
A_{2,1}^{n+1} & A_{2,2}^{n+1} & A_{2,3}^{n+1} & \cdots & A_{2,N}^{n+1} \\
A_{3,1}^{n+1} & A_{3,2}^{n+1} & A_{3,3}^{n+1} & \cdots & A_{3,N}^{n+1} \\
\vdots & \vdots & \vdots & \ddots & \vdots \\
A_{N,1}^{n+1} & A_{N,2}^{n+1} & A_{N,3}^{n+1} & \cdots & A_{N,N}^{n+1}
\end{bmatrix}
\begin{Bmatrix}
h_1^{n+1} \\ h_2^{n+1} \\ h_3^{n+1} \\ \vdots \\ h_N^{n+1}
\end{Bmatrix}
=
\begin{Bmatrix}
c_1^{n+1} \\ c_2^{n+1} \\ c_3^{n+1} \\ \vdots \\ c_N^{n+1}
\end{Bmatrix}
-
\begin{Bmatrix}
b_1^{n+1} \\ b_2^{n+1} \\ b_3^{n+1} \\ \vdots \\ b_N^{n+1}
\end{Bmatrix}. \tag{18}
$$

If three arbitrary nodes with node numbers $i_1$, $i_2$, and $i_3$ correspond to a Dirichlet boundary, and the Dirichlet boundary values at the $i_1$, $i_2$, and $i_3$ nodes are set to $h_{i1}$, $h_{i2}$, and $h_{i3}$, respectively, Equation (18) can be conventionally changed by setting the rows corresponding to node numbers $i_1$, $i_2$, and $i_3$ to zero, the diagonal terms to 1, and the corresponding rows of the load vector to the Dirichlet boundary values $h_{i1}$, $h_{i2}$, and $h_{i3}$, as follows:

$$
\begin{bmatrix}
A_{1,1}^{n+1} & A_{1,2}^{n+1} & \cdots & A_{1,i1}^{n+1} & \cdots & A_{1,i2}^{n+1} & \cdots & A_{1,i3}^{n+1} & \cdots & A_{1,N}^{n+1} \\
A_{2,1}^{n+1} & A_{2,2}^{n+1} & \cdots & A_{2,i1}^{n+1} & \cdots & A_{2,i2}^{n+1} & \cdots & A_{2,i3}^{n+1} & \cdots & A_{2,N}^{n+1} \\
\vdots & \vdots & \vdots & \vdots & \vdots & \vdots & \vdots & \vdots & \vdots & \vdots \\
0 & 0 & 0 & 1 & 0 & 0 & 0 & 0 & 0 & 0 \\
\vdots & \vdots & \vdots & \vdots & \vdots & \vdots & \vdots & \vdots & \vdots & \vdots \\
0 & 0 & 0 & 0 & 1 & 0 & 0 & 0 & 0 & 0 \\
\vdots & \vdots & \vdots & \vdots & \vdots & \vdots & \vdots & \vdots & \vdots & \vdots \\
0 & 0 & 0 & 0 & 0 & 1 & 0 & 0 & 0 & 0 \\
\vdots & \vdots & \vdots & \vdots & \vdots & \vdots & \vdots & \vdots & \vdots & \vdots \\
A_{N,1}^{n+1} & A_{N,2}^{n+1} & \cdots & A_{N,i1}^{n+1} & \cdots & A_{N,i2}^{n+1} & \cdots & A_{N,i3}^{n+1} & \cdots & A_{N,N}^{n+1}
\end{bmatrix}
\begin{Bmatrix}
h_1^{n+1} \\ h_2^{n+1} \\ \vdots \\ h_{i1}^{n+1} \\ \vdots \\ h_{i2}^{n+1} \\ \vdots \\ h_{i3}^{n+1} \\ \vdots \\ h_N^{n+1}
\end{Bmatrix}
=
\begin{Bmatrix}
c_1^{n+1} - b_1^{n+1} \\ c_2^{n+1} - b_2^{n+1} \\ \vdots \\ h_{i1} \\ \vdots \\ h_{i2} \\ \vdots \\ h_{i3} \\ \vdots \\ c_N^{n+1} - b_N^{n+1}
\end{Bmatrix}. \tag{19}
$$

Another conventional approach to accommodate the Dirichlet boundary condition is to simply discard rows and columns corresponding to Dirichlet boundaries such that the dimensions of the matrix and the load vector are reduced, and to modify the load terms at nodes connected to Dirichlet boundaries by moving the known Dirichlet boundary values to the right-hand side as follows:

$$
\begin{bmatrix}
A_{1,1}^{n+1} & A_{1,2}^{n+1} & \cdots & A_{1,N-3}^{n+1} \\
A_{2,1}^{n+1} & A_{2,2}^{n+1} & \cdots & A_{2,N-3}^{n+1} \\
\vdots & \vdots & \vdots & \vdots \\
A_{N-3,1}^{n+1} & A_{N-3,2}^{n+1} & \cdots & A_{N-3,N-3}^{n+1}
\end{bmatrix}
\begin{Bmatrix}
h_1^{n+1} \\ h_2^{n+1} \\ \vdots \\ h_{N-3}^{n+1}
\end{Bmatrix}
=
\begin{Bmatrix}
c_1^{n+1} - b_1^{n+1} - A_{1,1}^{n+1} h_{i1} - A_{1,i2}^{n+1} h_{i2} - A_{1,i3}^{n+1} h_{i3} \\
c_2^{n+1} - b_2^{n+1} - A_{2,1}^{n+1} h_{i1} - A_{2,i2}^{n+1} h_{i2} - A_{2,i3}^{n+1} h_{i3} \\
\vdots \\
c_{N-3}^{n+1} - b_{N-3}^{n+1} - A_{N-3,i1}^{n+1} h_{i1} - A_{N-3,i2}^{n+1} h_{i2} - A_{N-3,i3}^{n+1} h_{i3}
\end{Bmatrix}. \tag{20}
$$

The approach described in Equation (19) maintains the matrix size, whereas the approach described in Equation (20) reduces the matrix size by the number of Dirichlet boundary nodes, and hence the latter approach may be more computationally efficient, particularly for a large number of Dirichlet nodes. Approaches deriving Equations (19) or (20) after some manipulation are hereafter denoted as the typical conventional approach (TCA) for accommodating the Dirichlet boundary condition.

In contrast to the TCA above, the new approach maintains the Galerkin equations at the Dirichlet boundaries by setting the values of $\{b_i^{n+1}\}$, i.e., the Galerkin fluxes or integral boundary fluxes at the Dirichlet nodes, as unknown variables, and at the same time, moving the values of $\{h_i\}$, i.e., the a priori known hydraulic heads at the Dirichlet nodes, to the right-hand side. As an easier explanation, if the rows corresponding to node numbers $i_1$, $i_2$, and $i_3$ in Equation (18) are expressed in algebraic equations, the following equations can be obtained:

$$A_{i1,1}^{n+1}h_1^{n+1} + A_{i1,2}^{n+1}h_2^{n+1} + \cdots + A_{i1,i1}^{n+1}h_{i1}^{n+1} + \cdots + A_{i1,i2}^{n+1}h_{i2}^{n+1} + \cdots + A_{i1,i3}^{n+1}h_{i3}^{n+1} + \cdots + A_{i1,N}^{n+1}h_N^{n+1} = c_{i1}^{n+1} - b_{i1}^{n+1}, \quad (21)$$

$$A_{i2,1}^{n+1}h_1^{n+1} + A_{i2,2}^{n+1}h_2^{n+1} + \cdots + A_{i2,i1}^{n+1}h_{i1}^{n+1} + \cdots + A_{i2,i2}^{n+1}h_{i2}^{n+1} + \cdots + A_{i2,i3}^{n+1}h_{i3}^{n+1} + \cdots + A_{i2,N}^{n+1}h_N^{n+1} = c_{i2}^{n+1} - b_{i2}^{n+1}, \quad (22)$$

$$A_{i3,1}^{n+1}h_1^{n+1} + A_{i3,2}^{n+1}h_2^{n+1} + \cdots + A_{i3,i1}^{n+1}h_{i1}^{n+1} + \cdots + A_{i3,i2}^{n+1}h_{i2}^{n+1} + \cdots + A_{i3,i3}^{n+1}h_{i3}^{n+1} + \cdots + A_{i3,N}^{n+1}h_N^{n+1} = c_{i3}^{n+1} - b_{i3}^{n+1}. \quad (23)$$

Similarly, rows not corresponding to arbitrary node number $k$ that is not a Dirichlet node can be expressed as Equation (24):

$$A_{k,1}^{n+1}h_1^{n+1} + A_{k,2}^{n+1}h_2^{n+1} + \cdots + A_{k,i1}^{n+1}h_{i1}^{n+1} + \cdots + A_{k,i2}^{n+1}h_{i2}^{n+1} + \cdots + A_{k,i3}^{n+1}h_{i3}^{n+1} + \cdots + A_{k,N}^{n+1}h_N^{n+1} = c_k^{n+1} - b_k^{n+1}. \quad (24)$$

If the known values of $h_{i1}^{n+1}$, $h_{i2}^{n+1}$, and $h_{i3}^{n+1}$ (i.e., $h_{i1}$, $h_{i2}$, $h_{i3}$, respectively) are moved to the right-hand side, and the unknown values of $b_{i1}^{n+1}$, $b_{i2}^{n+1}$, and $b_{i3}^{n+1}$ are moved to the left-hand side, Equations (21)–(24) can be changed to Equations (25)–(28), respectively, as follows:

$$A_{i1,1}^{n+1}h_1^{n+1} + A_{i1,2}^{n+1}h_2^{n+1} + \cdots + b_{i1}^{n+1} + \cdots + 0 + \cdots + 0 + \cdots + A_{i1,N}^{n+1}h_N^{n+1} = c_{i1}^{n+1} - A_{i1,i1}^{n+1}h_{i1} - A_{i1,i2}^{n+1}h_{i2} - A_{i1,i3}^{n+1}h_{i3}, \quad (25)$$

$$A_{i2,1}^{n+1}h_1^{n+1} + A_{i2,2}^{n+1}h_2^{n+1} + \cdots + 0 + \cdots + b_{i2}^{n+1} + \cdots + 0 + \cdots + A_{i2,N}^{n+1}h_N^{n+1} = c_{i2}^{n+1} - A_{i2,i1}^{n+1}h_{i1} - A_{i2,i2}^{n+1}h_{i2} - A_{i2,i3}^{n+1}h_{i3}, \quad (26)$$

$$A_{i3,1}^{n+1}h_1^{n+1} + A_{i3,2}^{n+1}h_2^{n+1} + \cdots + 0 + \cdots + 0 + \cdots + b_{i3}^{n+1} + \cdots + A_{i3,N}^{n+1}h_N^{n+1} = c_{i3}^{n+1} - A_{i3,i1}^{n+1}h_{i1} - A_{i3,i2}^{n+1}h_{i2} - A_{i3,i3}^{n+1}h_{i3}, \quad (27)$$

$$A_{k,1}^{n+1}h_1^{n+1} + A_{k,2}^{n+1}h_2^{n+1} + \cdots + 0 + \cdots + 0 + \cdots + 0 + \cdots + A_{k,N}^{n+1}h_N^{n+1} = c_k^{n+1} - b_k^{n+1} - A_{k,i1}^{n+1}h_{i1}^{n+1} - A_{k,i2}^{n+1}h_{i2}^{n+1} - A_{k,i3}^{n+1}h_{i3}^{n+1}. \quad (28)$$

Finally, if the set of Equations (25)–(28) are expressed in a matrix form in the new approach, a new expression for the simultaneous algebraic equation system can be written as follows:

$$
\begin{bmatrix}
A_{1,1}^{n+1} & A_{1,2}^{n+1} & \cdots 0 & \cdots 0 & \cdots & 0 \cdots & A_{1,N}^{n+1} \\
A_{2,1}^{n+1} & A_{2,2}^{n+1} & \cdots 0 & \cdots 0 & \cdots & 0 \cdots & A_{2,N}^{n+1} \\
\vdots & \vdots & \cdots 0 & \cdots 0 & \cdots & 0 \cdots & \vdots \\
A_{i1,1}^{n+1} & A_{i1,2}^{n+1} & \cdots 1 & \cdots 0 & \cdots & 0 \cdots & A_{i1,N}^{n+1} \\
\vdots & \vdots & \cdots 0 & \cdots 0 & \cdots & 0 \cdots & \vdots \\
A_{i2,1}^{n+1} & A_{i2,2}^{n+1} & \cdots 0 & \cdots 1 & \cdots & 0 \cdots & A_{i2,N}^{n+1} \\
\vdots & \vdots & \cdots 0 & \cdots 0 & \cdots & 0 \cdots & \vdots \\
A_{i3,1}^{n+1} & A_{i3,2}^{n+1} & \cdots 0 & \cdots 0 & \cdots & 1 \cdots & A_{i3,N}^{n+1} \\
\vdots & \vdots & \cdots 0 & \cdots 0 & \cdots & 0 \cdots & \vdots \\
A_{N,1}^{n+1} & A_{N,2}^{n+1} & \cdots 0 & \cdots 0 & \cdots & 0 \cdots & A_{N,N}^{n+1}
\end{bmatrix}
\begin{Bmatrix}
h_1^{n+1} \\ h_2^{n+1} \\ \vdots \\ b_{i1}^{n+1} \\ \vdots \\ b_{i2}^{n+1} \\ \vdots \\ b_{i3}^{n+1} \\ \vdots \\ h_N^{n+1}
\end{Bmatrix}
=
\begin{Bmatrix}
c_1^{n+1} - b_1^{n+1} - A_{1,i1}^{n+1}h_{i1} - A_{1,i2}^{n+1}h_{i2} - A_{1,i3}^{n+1}h_{i3} \\
c_2^{n+1} - b_2^{n+1} - A_{2,i1}^{n+1}h_{i1} - A_{2,i2}^{n+1}h_{i2} - A_{2,i3}^{n+1}h_{i3} \\
\vdots \\
c_{i1}^{n+1} - A_{i1,i1}^{n+1}h_{i1} - A_{i1,i2}^{n+1}h_{i2} - A_{i1,i3}^{n+1}h_{i3} \\
\vdots \\
c_{i2}^{n+1} - A_{i2,i1}^{n+1}h_{i1} - A_{i2,i2}^{n+1}h_{i2} - A_{i2,13}^{n+1}h_{i3} \\
\vdots \\
c_{i3}^{n+1} - A_{i3,i1}^{n+1}h_{i1} - A_{i3,i2}^{n+1}h_{i2} - A_{i3,i3}^{n+1}h_{i3} \\
\vdots \\
c_N^{n+1} - b_N^{n+1} - A_{N,i1}^{n+1}h_{i1} - A_{N,i2}^{n+1}h_{i2} - A_{N,i3}^{n+1}h_{i3}
\end{Bmatrix}. \quad (29)
$$

As shown in Equation (29), the unknown variables to be solved are the hydraulic heads at all nodes, except the Dirichlet nodes, along with the Galerkin fluxes at the Dirichlet nodes. Therefore, in the new approach, by solving the simultaneous algebraic systems in Equation (29), the boundary fluxes at the

Dirichlet nodes and the finite element hydraulic head solutions at all nodes, except the Dirichlet nodes, can be obtained simultaneously in one step. Compared to the new approach, Yeh's approach and the CDA consist of two steps, wherein the first step is to obtain the Galerkin finite element hydraulic heads at all nodes by solving Equations (19) or (20), and then, calculate the boundary fluxes using the obtained Galerkin finite element hydraulic heads. Hence, the new approach presents a more straightforward and efficient computational procedure.

The global mass balance can be obtained by integrating Equation (1) over the whole space domain *R*, applying the divergence theorem, and substituting Equation (4), as follows:

$$\int_R F \frac{\partial h}{\partial t} dR - \int_R Q dR + \int_B \hat{n} \cdot V dB = 0. \tag{30}$$

The first and second terms on the left-hand side of Equation (30) represent, respectively, the volumetric rate of increase in moisture content and the mass change rate due to sinks/sources, with the latter being positive for withdrawal over the whole region *R*. The last term indicates the outwardly normal flux through the global boundary *B*. If we assume, for the sake of simplicity, that a global boundary consists of a Dirichlet boundary $B_d$, a Neumann boundary $B_n$, and an impermeable boundary $B_{im}$, the flux through the whole boundary can be divided into three components, as shown below:

$$\int_B \hat{n} \cdot V dB = \int_{B_d} \hat{n} \cdot V dB + \int_{B_n} \hat{n} \cdot V dB + \int_{B_{im}} \hat{n} \cdot V dB = F_d + F_n + F_{im}, \tag{31}$$

where $F_d$, $F_n$, and $F_{im}$ represent the fluxes through a Dirichlet boundary $B_d$, Neumann boundary $B_n$, and impermeable boundary $B_{im}$, respectively. To evaluate the mass balance computational performance of the different approaches, the global mass balance error over a whole region can be defined as:

$$\text{Mass balance error (MBE)} = \frac{\text{total net mass through all boundaries} - \text{mass change in region}}{\text{total mass accumulated in region}} \times 100. \tag{32}$$

If the total mass accumulated within a certain period $\Delta t$ in a region can be obtained as the sum of the total net mass through all boundaries during $\Delta t$ and the mass added (or removed) to the initial mass in the region during $\Delta t$ due to sources (or sinks), Equation (32) can be rewritten as:

$$\text{MBE} = \frac{-\int_B \hat{n} \cdot V dB \times \Delta t - \int_R F \frac{\partial h}{\partial t} dR \times \Delta t - \int_R Q dR \times \Delta t}{\int_R \theta(t_0) dR - \int_B \hat{n} \cdot V dB \times \Delta t + \int_R Q dR \times \Delta t} \times 100, \tag{33}$$

where $\theta(t_0)$ is the moisture content distribution at the initial time $t_0$. Here, at the Neumann and impermeable boundaries, respectively, $F_n$ and $F_{im}$ are known a priori, i.e., prescribed with known values. Accordingly, if the flux through a Dirichlet boundary can be calculated exactly and the finite element hydraulic solutions obtained from Equations (19), (20), or (29) are free of error, the mass balance equation (Equation (30)) will be perfectly satisfied, assuming that the numerical quadrature is exact and there is no temporal discretization error. The flux through the Dirichlet boundary can be conventionally calculated by either differentiating the hydraulic heads computed from Equation (19) or (20) using the CDA, or by applying the finite element approach to the Darcy equation with the hydraulic heads computed from Equation (19) or (20) using Yeh's approach. However, the approach proposed in this study directly calculates the flux through Dirichlet boundaries by solving Equation (29) for Galerkin fluxes or integral boundary fluxes at the Dirichlet nodes. To illustrate the superiority of the mass balance computation performed using the proposed approach, here, the approach was compared to the CDA and Yeh's approach using two practical groundwater scenarios.

### 3. Results and Discussion

Two hypothetical groundwater examples were used in this study to compare the proposed approach to the CDA and Yeh's approach. The first problem considered the case of infiltration through the bottom of a long ditch of a certain width, which is parallel to two long rivers that bound an unconfined aquifer. This example was adapted from Strack [24]. The second problem involved a hypothetical small watershed with a sloping area, as described by Yeh [4], with the exception of the boundary conditions. In the first example, two different discretization schemes were applied, with the whole domain discretized into fine and coarse grids, respectively. Within the two discretization systems, the accuracies of the different finite element hydraulic head solutions over the entire domain obtained through the TCA or new approach and the accuracies of the fluxes at Dirichlet boundaries obtained using each of the three computational approaches were compared to analytical solutions of the hydraulic head distribution and the boundary flux. Similarly, for the second example, the cumulative mass balance error was calculated and the CPU time was recorded for each of the three computational approaches for comparison.

### 3.1. Example 1: Steady-State Infiltration through the Bottom of a Long Ditch

In the first example, it was assumed that the bottom of the ditch was not in contact with the phreatic surface and that water will leak through the bottom, filter down to the phreatic surface, and finally join the flow in the aquifer. As shown in Figure 1, the distance between the rivers $L$ was 200 m and the heads along the left and right rivers were 1 and 3 m, respectively. The rate of steady infiltration through the bottom of the ditch was 0.2 m/day. The coordinates of the boundaries of the ditch were $\xi_1$ and $\xi_2$ at 10 and 20 m, respectively; hence, the width of the long ditch $b$ was 10 m. To calculate the hydraulic head distribution over the whole domain and the fluxes at the boundary, the whole domain was discretized with rectangular elements at each of two different discretization levels, as shown in Figure 2. For the fine and coarse discretization, the largest element sizes in the $x$-direction were 1 and 20 m, respectively. To consider the rate of infiltration through the bottom of the ditch, a Neumann boundary condition was specified along the ditch; Dirichlet boundary conditions were assigned along the rivers, as shown in Figure 2. All other boundaries were assumed to be impermeable. Modeling of the unsaturated flow in the zone between the bottom of the ditch and the phreatic surface was not conducted to keep the numerical solutions of the hydraulic head distribution and boundary flux consistent with the analytical solutions developed by Strack [24], in which the unsaturated flow was not considered. Accordingly, to account for the occurrence of unsaturated flow during the simulation, the curves of the constitutive relations that associate the pressure head with the water content and relative permeability were determined, as shown in Figure 3, where the material properties of the constitutive relations are shown in Table 1. In this constitutive relation, when the pressure head was less than 0, the water content became $\theta_r$, and thus, the relative permeability was 0. The steady-state hydraulic head distributions were calculated through Equations (19), (20), and (29) using the TCA and the new approach, and subsequently, the velocity field distributions in the whole domain were obtained through postprocessing using either Yeh's approach or the CDA at all different discretization levels to compute the integral boundary fluxes for the mass balance computation.

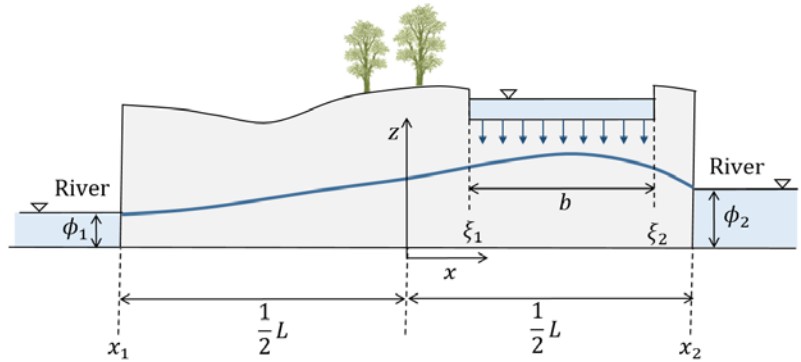

**Figure 1.** Schematic diagram showing local infiltration through the bottom of a long ditch (modified from Strack [24]). $\phi_1$ and $\phi_2$ are the heads along the left and right rivers, respectively; $L$ is the distance between the rivers; $x_1$ and $x_2$ are the coordinates of the left- and right-hand boundaries of the interested aquifer, respectively; $\xi_1$ and $\xi_2$ are the coordinates of the left- and right-hand boundaries of the ditch, respectively; $b$ is the width of the long ditch; and $z$ is the elevation.

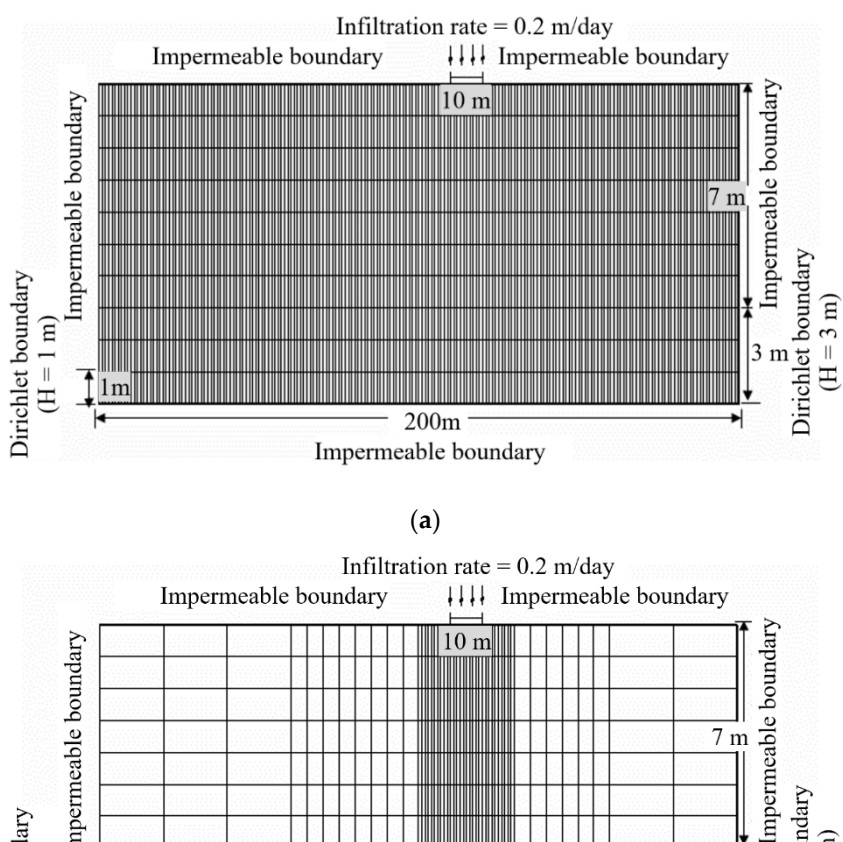

**Figure 2.** Discretized domains with (**a**) fine and (**b**) coarse rectangular elements and boundary conditions.

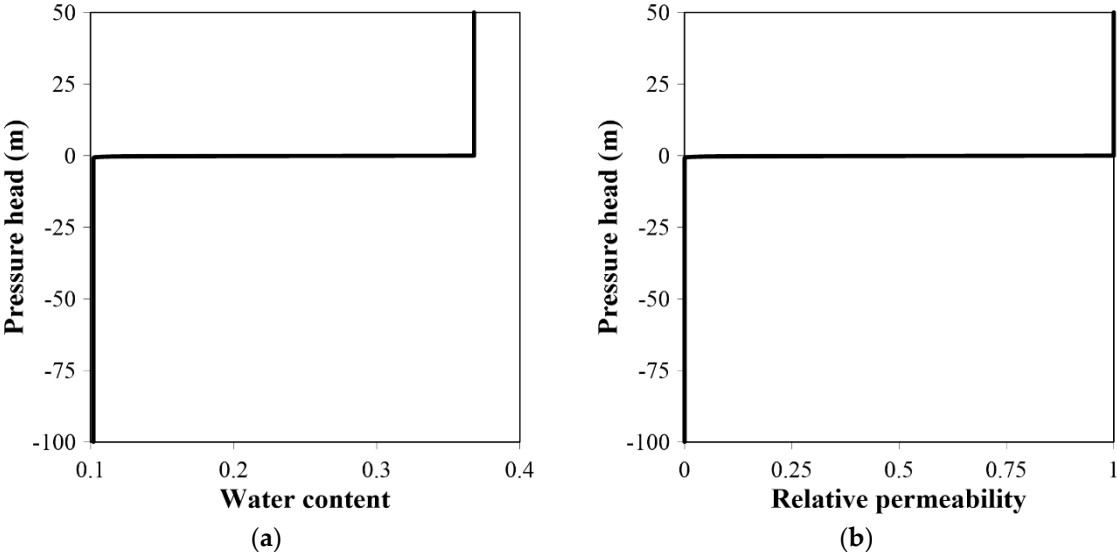

**Figure 3.** Curves of constitutive relations associating the pressure head with (**a**) the water content and (**b**) the relative permeability.

**Table 1.** Soil properties determining the curves of the constitutive relations. $K$, $\theta_s$, $\theta_r$, and $\lambda$ are the hydraulic conductivity, saturated water content, relative water content, and soil index parameter related to the pore-size distribution, respectively.

| Soil Properties | Example 1 | Example 2 |
|---|---|---|
| K (m/day) | 10 | 0.01 |
| $\theta_s$ (m³/m³) | 0.368 | 0.368 |
| $\theta_r$ (m³/m³) | 0.102 | 0.102 |
| $\lambda$ (m⁻¹) | 9.8 | 1.0 |

As shown in Figure 4, the difference between the velocity field distributions of Yeh's approach and the CDA seemed to be small by visual inspection alone, except around the upconing of the water table beneath the ditch, where the velocity obtained using the CDA was significantly larger than that using Yeh's approach. However, to provide a more sophisticated analysis, the difference between velocity field distributions calculated using the Yeh's postprocessing approach and the CDA method were quantified by calculating the root mean square (RMS) of the normalized velocity difference vector. The root mean square (RMS) of the normalized velocity difference vector was defined as follows:

$$\text{normalized RMS} = \sqrt{\frac{1}{N}\sum_{i=1}^{N}\left(\frac{V_i^{Yeh} - V_i^{CDA}}{V_{i,n}^{Yeh}}\right)^2}, \quad i \in N,$$

$$V_{i,n}^{Yeh} = \begin{cases} V_i^{Yeh}, & if\ V_i^{Yeh} \geq V_{max}^{Yeh} \times 0.001, \\ V_{max}^{Yeh} \times 0.001, & if\ V_i^{Yeh} < V_{max}^{Yeh} \times 0.001, \end{cases} \tag{34}$$

where $N$ is the total number of nodes; $V_i^{Yeh}$ and $V_i^{CDA}$ are the magnitudes of velocities at the *i*th node calculated using the Yeh's approach and CDA, respectively; and $V_{max}^{Yeh}$ is the maximum value of $V_i^{Yeh}$ ($i = 1, 2, \ldots, N$). The RMS values of the normalized velocity differences calculated according to Equation (34) are indicated in Table 2.

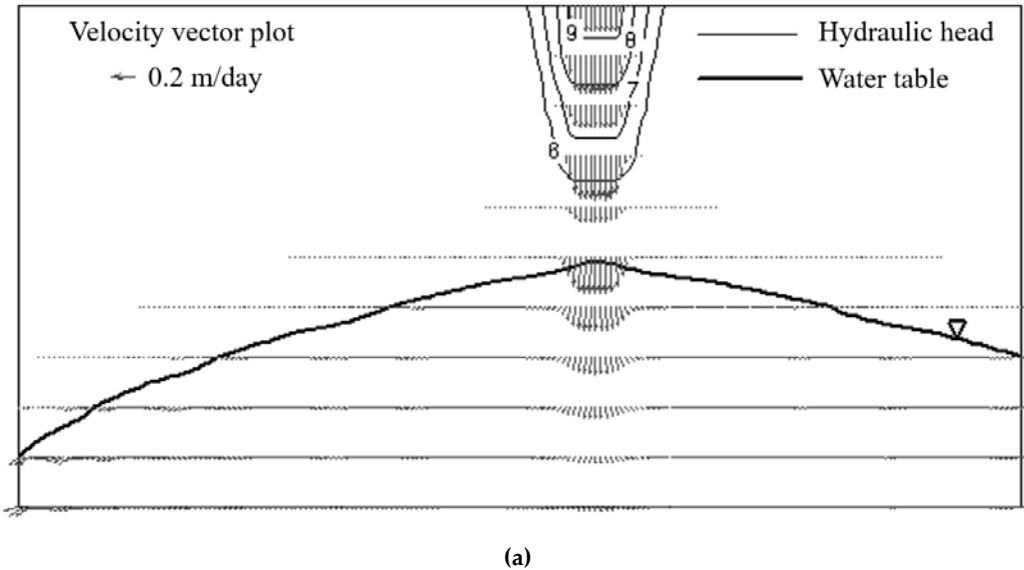

**(a)**

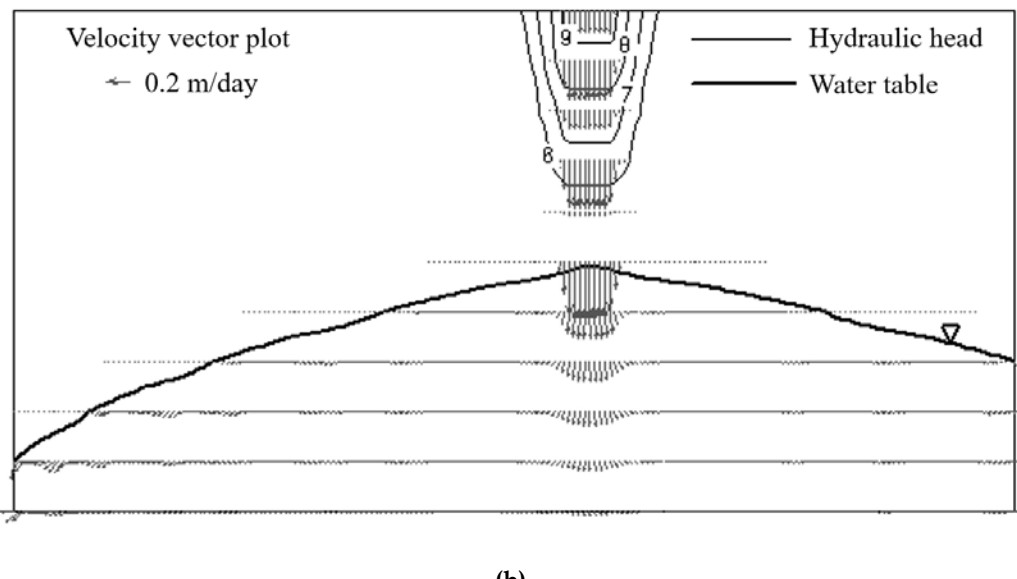

**(b)**

**Figure 4.** Steady-state hydraulic heads calculated using the typical conventional approach (TCA) and velocity field distributions obtained using (**a**) Yeh's postprocessing approach and (**b**) the conventional differential approach (CDA) at the fine discretization level.

**Table 2.** The RMS values of the normalized velocity differences calculated using Equation (34).

| Spatial Discretization Level | Normalized RMS (%) |
| --- | --- |
| Fine | 32.9 |
| Coarse | 33.6 |

As shown in Table 2, the RMS values of the normalized velocity differences were 32.9% and 33.6% in the fine and coarse discretization levels, respectively, which are not small. It was found out that these results are reasonably consistent with those found in Yeh [4].

The analytical solution from Strack [24] was adapted for a more accurate comparison, as follows:

$$\Phi(x) = \sqrt{\frac{2}{K}\left[-\Phi_1\frac{(x-0.5L)}{L} + \Phi_2\frac{(x+0.5L)}{L} + (\hat{n}\cdot V)_{BC}\cdot G_d(x)\right]} \tag{35}$$

where $\Phi(x)$ is the elevation of the water table at $x$; $\Phi_1$ and $\Phi_2$ are the discharge potentials for the horizontal flow at the left- and right-hand boundaries, respectively; $(\hat{n}\cdot V)_{BC}$ is the rate of infiltration through the bottom of the ditch; and $L$ is the width of the aquifer. Here, $\Phi_1 = \frac{1}{2}K\phi_1^2$ and $\Phi_2 = \frac{1}{2}K\phi_2^2$, where $\phi_1$ and $\phi_2$ are the elevations of the water table at the left- and right-hand boundaries, respectively. $G_d(x)$ from Equation (35) was calculated as follows:

$$G_d(x) = \frac{(x+0.5L)}{L}\left[(0.5L-\xi_2)b + 0.5b^2\right], \qquad x_1 \leq x \leq \xi_1, \tag{36}$$

$$G_d(x) = \frac{(x+0.5L)}{L}\left[(0.5L-\xi_2)b + 0.5b^2\right] - 0.5(x-\xi_1)^2, \qquad \xi_1 \leq x \leq \xi_2, \tag{37}$$

$$G_d(x) = \frac{(x+0.5L)}{L}\left[(0.5L-\xi_2)b + 0.5b^2\right] - b(x-\xi_2) - 0.5b^2, \qquad \xi_2 \leq x \leq x_2, \tag{38}$$

where $b$ is the width of the ditch and $b = \xi_2 - \xi_1$. In addition, the analytical discharges at the left and right Dirichlet boundaries could be obtained by differentiating Equation (35) with respect to $x$ and calculating the discharges at $x_1$ and $x_2$, as follows:

$$\left.\frac{\partial\Phi}{\partial x}\right|_{x=x_1} = \left\{-\frac{\Phi_1}{L} + \frac{\Phi_2}{L} + (\hat{n}\cdot V)_{BC}\cdot\frac{\left[(0.5L-\xi_2)b + 0.5b^2\right]}{L}\right\}, \tag{39}$$

$$\left.\frac{\partial\Phi}{\partial x}\right|_{x=x_2} = \left\{-\frac{\Phi_1}{L} + \frac{\Phi_2}{L} + (\hat{n}\cdot V)_{BC}\cdot\frac{\left[(0.5L-\xi_2)b + 0.5b^2 - bL\right]}{L}\right\}. \tag{40}$$

The analytical solutions allowed for the accuracy of the computed water table profiles obtained through the new approach for solving Equation (29) and through the TCA for solving Equations (19) or (20) under the two different discretization levels to be evaluated in greater detail. Furthermore, as shown in Figure 5, the computed water table profiles could be plotted against the exact analytical solutions from Equation (35). The maximum difference $E_h$ between the exact analytical and numerical solutions was calculated thus:

$$E_h = max\left|\hat{\phi}(x_i) - \phi(x_i)\right|, \qquad i \in N, \tag{41}$$

where $\hat{\phi}(x_i)$ is the computed water table at location $x_i$ and $\phi(x_i)$ is the analytical solution of Equation (35) at location $x_i$. According to Figure 5, the water table profiles computed using the new method to solve Equation (29) and using the TCA to solve Equations (19) or (20) were identical. Indeed, we found good agreement between the two results, regardless of the mesh discretization level (not shown). However, the maximum difference $E_h$ increased with increasing element size in both approaches, as shown in Table 3.

**Table 3.** The maximum difference $E_h$ between the exact analytical and numerical solutions obtained through the new approach to solve Equation (29) and the TCA to solve Equation (19) or (20) at two spatial discretization levels.

| Spatial Discretization Level | New Approach's $E_h$ (m) | TCA's $E_h$ (m) |
|:---:|:---:|:---:|
| Fine | $6.16 \times 10^{-2}$ | $6.16 \times 10^{-2}$ |
| Coarse | $8.55 \times 10^{-2}$ | $8.55 \times 10^{-2}$ |

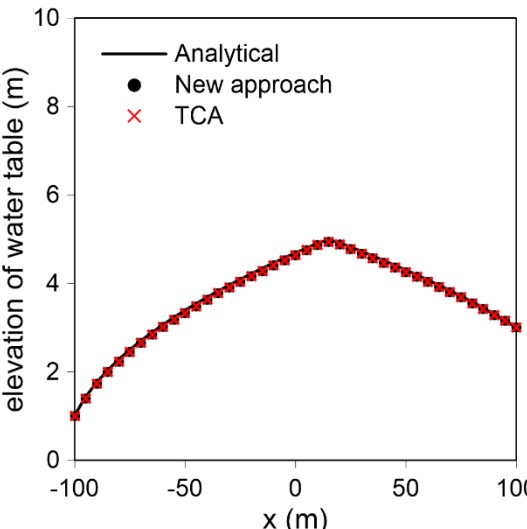

**Figure 5.** Plot of the computed water table profiles obtained through the new approach to solve Equation (29) and through the TCA to solve Equations (19) or (20) compared with the exact analytical solution at the fine discretization level.

In addition to the computed water table profiles, $E_f$, which is the percentage error between the exact analytical solution and the integral boundary flux at the Dirichlet boundaries computed using the three methods, was calculated as follows:

$$E_f = \frac{(F_A - F_N)}{F_A} \times 100, \tag{42}$$

where $F_A$ is the analytical boundary discharge obtained using Equations (39) and (40) and $F_N$ is the boundary flux at the Dirichlet boundaries obtained through Yeh's approach, the CDA, or the new approach. According to Table 4, at the fine discretization level, the percentage errors between the exact analytical solutions and the computed boundary fluxes on the left- and right-hand sides using the new approach were 0.96% and 1.07%, respectively, whereas those between the exact analytical solutions and the computed boundary fluxes on the left- and right-hand sides using Yeh's approach were 1.06% and 1.59%, respectively. Yeh's approach produced slightly larger errors (1.1–1.5-fold) than the new approach. Furthermore, in the CDA, the errors on the left- and right-hand sides were 8.55% and 3.23%, respectively, which were larger than those produced by the new method by 3.0 and 8.9 times, respectively. Similarly, at the coarse discretization level, the errors on the left- and right-hand sides produced using the new approach were 1.63% and 1.81%, respectively, and those on the left- and right-hand sides using Yeh's approach were 19.23% and 6.16%, respectively. At the coarse discretization level, Yeh's approach produced much larger errors than the new approach. Moreover, in the CDA, as the mesh became coarser, the errors on the left- and right-hand sides increased drastically from 8.55% and 3.23% to 42.61% and 12.30%, respectively. Therefore, regardless of the mesh discretization level, the new approach produced the most accurate results in the calculation of boundary fluxes, which is a requisite for mass balance computations. In addition, the CDA yielded larger errors compared to Yeh's approach, which is consistent with previous findings.

**Table 4.** Boundary fluxes at the Dirichlet boundaries computed using three methods: Yeh's approach, the CDA, and the new approach, under two different discretization levels. The exact analytical solutions are also shown, and the percentage errors between the exact analytical solutions and the computed boundary fluxes are given in parentheses.

| Methods | Boundary Sides | Boundary Fluxes at the Dirichlet Boundary ($m^3$/day) under Different Discretization Levels | |
|---|---|---|---|
| Analytical solution | Left/right | 1.05/0.95 | |
| | | Fine | Coarse |
| New approach | Left/right | 1.060/0.940 ($E_f$ = 0.96%/1.07%) | 1.067/0.933 (1.63%/1.81%) |
| Yeh's approach | Left/right | 1.039/0.935 (1.06%/1.59%) | 0.881/0.895 (19.23%/6.16%) |
| CDA | Left/right | 0.960/0.919 (8.55%/3.23%) | 0.603/0.833 (42.61%/12.30%) |

### 3.2. Example 2: A Problem Involving a Hypothetical Small Watershed with a Sloping Area

In this section, a problem involving a hypothetical small watershed with a sloping area described in Yeh [4], with the exception of the boundary conditions, was computed to compare the efficiency and accuracy of the new approach with Yeh's approach and the CDA. The aquifer system was assumed to be composed of homogeneous sand with the unsaturated properties given in Table 1. It was assumed that streams flowed adjacent to the left and right boundaries of the aquifer. For the finite element calculation, the entire region was discretized with three-dimensional hexahedral elements. The total numbers of nodes and elements were 806 and 360, respectively. Figure 6 shows a vertical cross-section of the discretized domain. Nine elements on the top plateau surface were considered constant Neumann flux elements, as shown in Figure 6, and were assigned a constant infiltration rate of 0.1 cm/day. The sloping sides, along with the bottom, front, and back sides, were considered impermeable boundaries (lower boundary surfaces *A–E* and upper boundary surfaces *F–I*). As shown in Figure 6, on the interfaces between the aquifer and the streams, Dirichlet boundary conditions were designated and the hydraulic heads at the left and right boundaries were set at 130 and 190 cm, respectively. To obtain the initial conditions for a transient simulation, a constant infiltration rate of 0 cm/day was set at the top plateau surface. A variable time step size was used, with an initial time step size of $10^{-5}$ days, and each subsequent time step size increased 1.1-fold with a maximum time step size not greater than 0.1 days. The hydraulic head distributions were obtained using the new approach and the TCA over a total simulation time of 100 days. Comparisons of the hydraulic head distributions over a simulation time of 100 days obtained using the new approach and the TCA are shown in Figure 7, where the hydraulic head distributions were identical. Because there were no available analytical solutions, the numerical results of the hydraulic head distributions were compared against those obtained with a very fine discretization system; it was found that the numerical solutions were very similar at this fine discretization level (not shown). The MBEs of Equation (33) calculated through the new approach, Yeh's approach, and the CDA, as well as the CPU times for all approaches, are compared in Table 5. The new approach had faster CPU times (by 31.8% and 32.4%, respectively) compared to the CDA and Yeh's approach, and the MBE of the new approach was less than those of the CDA and Yeh's approach by approximately 950-fold and 552-fold, respectively. The reason for this was that, first, the new approach took only one step to obtain the boundary fluxes at the Dirichlet nodes to be used in the mass balance computation, whereas the CDA and Yeh's approach required a second step for solving the velocity fields. Therefore, the CDA and Yeh's approach needed a longer computational time to calculate the velocity fields in the additional second step. Moreover, the new approach was much more accurate than Yeh's approach or the CDA because the latter two approaches calculated the boundary fluxes incorrectly. In the CDA, a discontinuous velocity field is generated at nodal points and element boundaries, leading to a significant boundary flux error. Although it has been reported

that Yeh's approach produces much better results than the CDA by obtaining continuous velocity fields [4–6,8–10], in this example, Yeh's approach produced significant errors comparable to those from the CDA because of the large boundary flux errors along the impermeable boundary in the complicated flow regime. The boundary flux should theoretically be zero along an impermeable boundary; however, the velocity obtained through Yeh's approach did not satisfy this theory, as discussed in previous studies [4]. In particular, the boundary flux errors could be severe at an impermeable boundary zone within the convergent region of the flow (boundary surface segments *FG*, *BC,* and *CD*) in Figure 8, where the magnitude and direction of the flow velocity changed dramatically, as shown in Figure 7. Accordingly, even using Yeh's approach, significant global mass balance errors can occur when an impermeable boundary is located in a highly complicated flow regime, as shown in Figure 8. The MBE calculations at different numerical simulation times were performed using the new approach, as well as Yeh's approach and the CDA, which are shown in Figure 9. Compared to the new approach, the MBEs obtained through the other approaches increased with time because the boundary fluxes through an impermeable boundary were always calculated to be nonzero at all time steps and accumulated over time in the MBE calculations. These erroneous nonzero boundary fluxes originated from the inaccurate velocity calculations in Yeh's approach and the CDA at these boundaries. Although these errors at the impermeable boundaries can be reduced by refining elements in the CDA and Yeh's approach, the new approach at this discretization level still produced mass balance errors within only approximately $3.37 \times 10^{-3}\%$ for all simulation times, as shown in Table 5. Therefore, in these typical groundwater scenarios, the new approach provided a much more accurate MBE calculation compared to the other approaches for all simulation times. The new approach also demonstrated superior efficiency.

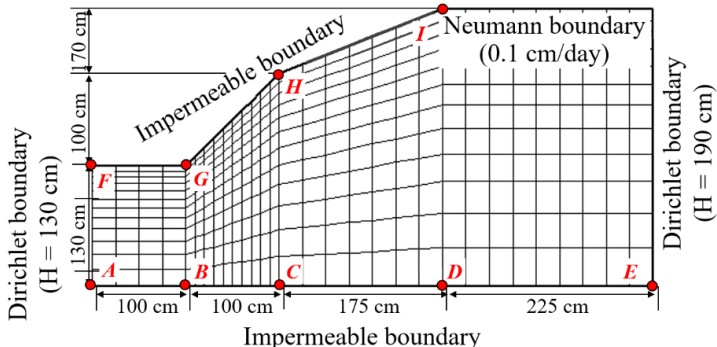

**Figure 6.** Configuration of a problem involving a hypothetical small watershed with a sloping area showing spatial finite element discretization, boundary conditions, and segments of impermeable boundary surfaces (*A–E* for the lower boundary and *F–I* for the upper boundary).

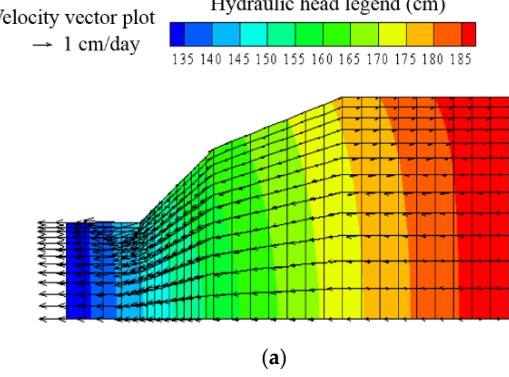

(**a**)

**Figure 7.** *Cont.*

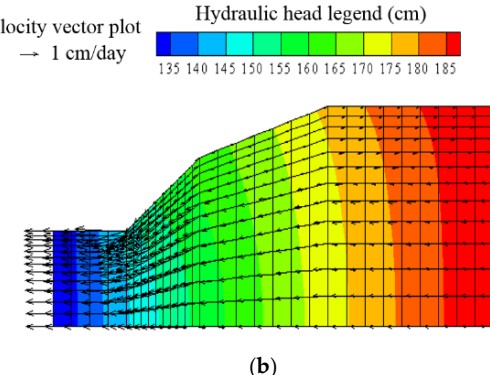

(**b**)

**Figure 7.** Comparison of the hydraulic head distributions obtained through the (**a**) TCA and (**b**) new approach and comparison of the velocity field distributions obtained via (**a**) the CDA and (**b**) Yeh's approach.

**Table 5.** Cumulative global mass balance errors of Equation (33) calculated for the new, CDA, and Yeh's approaches, as well as the CPU times for a simulation time of 100 days.

| Methods | CPU (s) | Mass Balance Error (%) |
|---|---|---|
| New approach | 81.35 | $3.37 \times 10^{-3}$ |
| CDA | 119.36 | 3.20 |
| Yeh's approach | 120.38 | 1.86 |

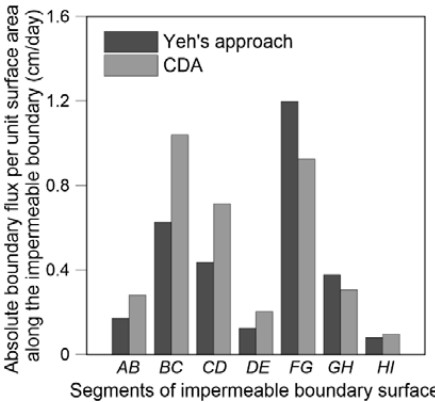

**Figure 8.** Absolute boundary flux errors per unit surface area along the impermeable boundary surface segments.

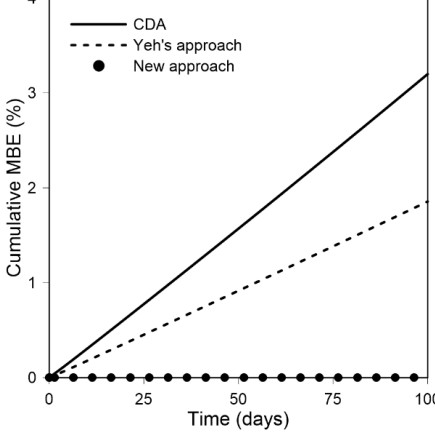

**Figure 9.** Comparison of cumulative mass balance errors (MBEs) as a function of the time calculated for the new approach, Yeh's approach, and CDA.

## 4. Conclusions

In this study, a simple new computational procedure based on the approach described by Lynch [11] was proposed to simultaneously obtain boundary fluxes at Dirichlet boundary nodes and finite element hydraulic heads at all nodes, except Dirichlet boundary nodes, within a single step to provide accurate mass balance computations. Compared to previous mass balance computational procedures, such as Yeh's approach and the CDA, which usually require two steps, the new approach was computationally more efficient and convenient. Most previous mass balance studies based on Lynch's approach [11] are limited in application to only simple mathematical concepts or hypothetical abstract examples that have a one-dimensional steady state with homogeneous material and simple boundary conditions or simple geometry. These examples are far from typical groundwater scenarios or realistic application scenarios. Accordingly, the proposed procedure was applied here to two typical groundwater scenarios. The first considered a case of infiltration through the bottom of a long ditch of a certain width, as adapted from Strack [24], and the second was a problem involving a hypothetical small watershed with a sloping area, as described in Yeh [4], with different boundary conditions and aquifer properties. In the first example, for two different spatial discretization levels, solutions derived using the proposed approach and two previous approaches (Yeh's approach and the CDA) were compared in terms of the accuracy of the calculated fluxes at the Dirichlet boundaries using analytical solutions. As the spatial discretization became coarser, the calculated maximum difference between the exact analytical solution and the numerical solutions computed through Yeh's approach and the CDA were much larger than the difference observed with the new approach. The calculated mass balance errors from the previous approaches increased significantly as the mesh became coarser; however, using the new approach, the errors increased only slightly to within approximately 2% at all mesh discretization levels.

Similarly, in the second example, the mass balance error of the new approach was much less (552-fold and 950-fold, respectively) than those of the previous approaches because Yeh's approach and the CDA yielded significant errors when calculating the velocities at the impermeable boundaries. Although it has been reported that Yeh's approach produces much better results than the CDA by obtaining continuous velocity fields, even Yeh's approach in this example produced significant errors when calculating the velocities, especially when an impermeable boundary was located in a highly complicated flow regime, leading to significant global mass balance errors. Furthermore, the CPU time of the new approach was approximately 32.4% and 31.8% faster than those of Yeh's approach and the CDA, respectively, because the new approach used only one step to obtain boundary fluxes at the Dirichlet nodes, whereas both Yeh's approach and the CDA needed a second step to compute the velocity fields. From the results of these numerical experiments, it can be concluded that the new approach provided more accurate and efficient mass balance computations compared to the previous approaches that are widely used in commercial and public groundwater software.

**Author Contributions:** Conceptualization, J.-S.C. and H.S.; methodology, H.S.; validation, H.S.; code development, H.S.; formal analysis, E.P.; investigation, E.P.; resources and facility, J.-S.C.; writing—original draft preparation, H.S..; writing—review and editing, J.-S.C. and Y.H.K.; visualization, H.S.; supervision, J.-S.C.; project administration, Y.H.K.; funding acquisition, Y.H.K. All authors have read and agreed to the published version of the manuscript.

**Funding:** This study was supported by the project titled "Research on rock properties in deep environment for HLW geological disposal (GP2020-002; 20-3115)" funded by the Ministry of Science and ICT, Korea.

**Acknowledgments:** This study was performed through an international collaborative research program between the National Central University (NCU) in Taiwan and the Korea Institute of Geosciences and Mineral resources (KIGAM). We also appreciate support by the project titled "Research on rock properties in deep environment for HLW geological disposal (GP2020-002; 20-3115)" funded by the Ministry of Science and ICT, Korea.

**Conflicts of Interest:** The authors declare no conflict of interest. The funders had no role in the design of the study; in the collection, analyses, or interpretation of data; in the writing of the manuscript; or in the decision to publish the results.

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
