# Peer review of "Practical Application of the Galerkin Finite Element Method with a Mass Conservation Scheme under Dirichlet Boundary Conditions to Solve Groundwater Problems"

_sustainability, doi:10.3390/su12145627_

Round 1

Reviewer 1 Report

Dear Authors,

Very well written paper. Fairly enjoy reading it. 

Does the error increase with increasing model size and/or complexity?

The only suggestion I have is to breakdown the step leading to equation 15. I find the explanation from lines 197 to 201 isn't sufficient enough to help readers to quickly derive equation 15.

Looking toward your reply. 

Thanks and best regards,

Eric

Author Response

Prof. Jui-Sheng Chen

Graduate Institute of Applied Geology,

National Central University, Taoyuan City 320, Taiwan

Tel: +886-3-2807427,

Dear whom concerned:

Please find attached files for a manuscript entitled Practical application of the Galerkin finite element method with a mass conservation scheme under Dirichlet boundary conditions to solve groundwater problems by Heejun Suk, Jui-Sheng Chen, Eungyu Park, and You Hong Kihm to be resubmitted as a revised paper to the Sustainability as an Article (manuscript no: sustainability-855346). Heejun Suk, Jui-Sheng Chen, Eungyu Park, and You Hong Kihm declare that all individuals listed as authors qualify as authors and have approved the submitted version; that the work is original and is not under consideration by any other journal.

Sincerely yours,

Jui-Sheng Chen

Reviewer 2 Report

Dear Authors, 

The paper entitled "Practical Application of the Galerkin Finite Element Method With a Mass Conservation Scheme Under Dirichlet Boundary Conditions to Solve Groundwater Problems" is a study based on a rigorous and robust methodology. The conclusions are correct and well documented, and the overall approach is consistent. 

It is a high quality work and does not need major changes.

In any case, it is important to make small modifications so that the study achieves greater coherence. Thus, a minor revision is proposed. 

The following changes are exposed:

  • It is important to include more bibliographic references to contextualize the study. One possibility is to write a longer Introduction, or even add a new section (2. Theoretical framework) between the Introduction and the Methodology.
  • The Results section should be called "Results and discussion", as the authors discuss the results as they are presented and then present the conclusions.

Authors are encouraged to correct these deficiencies for a high-quality paper. 

Best regards,

Reviewer

Author Response

(The authors gave the same response as above.)

Reviewer 3 Report

The manuscript details a one-step method for the solution of the groundwater flow equations, where previously two steps were used. The details of the method are first presented followed by two examples to illustrate its use. The manuscript is well written and can be published after some minor improvements to the presentation.

  1. It was hard to follow the geometry of example one. May be marking the rivers on Figure 1. Also, the way ξ 1 and ξ 2 are marked on the figure it looks like they are variables covering the distance, may just specify them at the points they represent, the way x1 and x2 are specified.

  1. Lines 294-297: “Because the difference between the velocity field distributions obtained using Yeh’s approach and the CDA cannot be discerned from visual inspection alone, only the velocity field distributions obtained using Yeh’s approach are shown in Figure 4, to demonstrate that the equations were reasonably solved.”   A discussion needs to be provided as to what these differences were.

Author Response

(The authors gave the same response as above.)
